# Unsupervised Analysis of Alzheimer's Disease Signatures using 3D Deformable Autoencoders

Anonymous

***@*.*

**Abstract.** With the increasing incidence of neurodegenerative diseases such as Alzheimer's Disease (AD), there is a need for further research that enhances detection and monitoring of the diseases. We present *MORPHADE* (Morphological Autoencoders for Alzheimer's Disease Detection), a novel unsupervised learning approach which uses deformations to allow the analysis of 3D T1-weighted brain images. To the best of our knowledge, this is the first use of deformations with deep unsupervised learning to not only detect, but also localize and assess the severity of structural changes in the brain due to AD. We obtain markedly higher anomaly scores in clinically important areas of the brain in subjects with AD compared to healthy controls, showcasing that our method is able to effectively locate AD-related atrophy. We additionally observe a visual correlation between the severity of atrophy highlighted in our anomaly maps and medial temporal lobe atrophy scores evaluated by a clinical expert. Finally, our method achieves an AUROC of 0.80 in detecting AD, out-performing several supervised and unsupervised baselines. We believe our framework shows promise as a tool towards improved understanding, monitoring and detection of AD. To support further research and application, we have made our code publicly available at https://anonymous.4open.science/r/MORPHADE-1925/.

**Keywords:** Unsupervised learning · Registration · Classification

## 1 Introduction

Due to the increased prevalence of neurodegenerative diseases and their effects on cognitive function, the study of such diseases is a highly active research field. As the leading cause of dementia [1], Alzheimer's disease (AD) is a particular focus of research advancements. However, the complex pathogenesis and progression mechanisms of AD remain only partially understood.

Magnetic resonance imaging (MRI) has shown use in the non-invasive tracking of AD-associated brain changes, such as hippocampal and amygdala atrophy and ventricular dilation [16,11]. Notably, several supervised machine learning methods utilizing MRI have been proposed which yield improvements in AD identification [24,15,23]. However, such methods are restricted by the need for large, annotated data sets. In contrast, unsupervised anomaly detection techniques [7,2,21,25] offer a promising solution by modeling the distribution of

healthy brain images to identify and localize anomalies without relying on labeled data.

Nevertheless, unsupervised approaches face challenges in accurately analyzing structural abnormalities, particularly regions of atrophy, which are critical in AD research [5]. Classical techniques using multi atlas-based deformable registration [13] and morphometry methods [8,3] have been proposed to analyze these structural changes. However, such methods allow analysis to be conducted only on a population-level, for instance as deviations from an atlas.

In this work, we propose Morphological Autoencoders for Alzheimer's Disease Detection ($MORPHADE$), a novel unsupervised anomaly detection framework based on deformable autoencoders (AEs) [4] which leverages deformation networks to generate patient-specific anomaly maps from 3D T1-weighted MRI brain scans. These anomaly maps allow not only AD detection, but also crucially reveal the location and degree of atrophy. Our main contributions are as follows:

- We use deformation fields in an unsupervised framework to analyze AD-related changes in the brain. To the best of our knowledge, this is the first use of such an approach using deep learning in the context of AD.
- We extend deformable autoencoders to 3D, utilize adversarial training and propose a dual-deformation strategy to improve reconstruction fidelity and the localization of atrophy.
- We accurately identify AD-affected brain regions, aligning our findings with clinical expectations.
- We assess AD severity by correlating our findings with clinical medial temporal lobe atrophy scores, evaluated by a board-certified clinical expert.
- Through comprehensive validation, we demonstrate superior performance in AD detection compared to unsupervised and even supervised baselines.

## 2   Background

In unsupervised anomaly detection, reconstruction-based frameworks such as autoencoders (AEs) can be used to learn the distribution of healthy samples and subsequently identify samples that deviate from this norm as anomalous. The encoder $E_\theta$ maps an input $x$ to a lower-dimensional latent space and then the decoder $D_\phi$ learns to reconstruct from this encoded representation. The parameters $\theta$, $\phi$ of the AE are optimized given healthy input data $\chi = \{x_i, ..., x_n\}$ by minimizing the mean squared error (MSE) between the inputs and their reconstructions:

$$MSE = min_{\theta,\phi} \sum_{i=1}^{N} ||x_i - D_\phi(E_\theta(x_i))||^2 \ . \tag{1}$$

It is then assumed that during inference, the AE will generate a so-called pseudo-healthy reconstruction, in which only in-distribution healthy tissue can be successfully reconstructed and thus any reconstruction errors can be thought

of as anomalies. A subject-specific map of anomalies can then be obtained by taking the residual between an input $x$ and its reconstruction $x_{recon} = D_\phi(E_\theta(x))$ as follows:

$$m_{residual} = |x - x_{recon}| \ . \tag{2}$$

Deformable Autoencoders (AEs) [4] were proposed as a method to alleviate false positives in the anomaly maps due to the limited reconstruction capabilities of traditional AEs. Since the top layers of the AE contain spatial information, deformable AEs use these layers to estimate a dense deformation field $\boldsymbol{\Phi}$ that allows local adaptions of the pseudo-healthy reconstruction to the individual anatomy of the subject. The estimation of the deformation field is optimized using local normalized cross correlation (LNCC):

$$\mathcal{L}_{morph} = LNCC(x, x_{morph}) + \beta||\boldsymbol{\Phi}||^2 \ , \tag{3}$$

where $\beta$ is a weight that is kept relatively high to constrain the deformations to be smooth and local, allowing only small changes to the reconstructions. We therefore refer to this part of the network as the constrained deformer. The improved reconstruction, which we refer to as the morphed reconstruction, $x_{morph}$, can then be obtained by $x_{morph} = x_{recon} \circ \boldsymbol{\Phi}$.

The authors also propose to use perceptual loss (PL) [12] weighted by the hyperparameter $\alpha$, in addition to the MSE when optimizing the AE parameters, to promote reconstructions that closely resemble the training distribution:

$$\mathcal{L}_{recon} = \text{MSE}(x, x_{recon}) + \alpha\text{PL}(x, x_{recon}) \ . \tag{4}$$

## 3   Methods and Materials

We propose *MORPHADE*, shown in Fig. 1, which builds upon deformable AEs. Firstly, we employ a 3D convolutional AE to enable the use of 3D images with the framework. Secondly, since PL uses 2D networks pre-trained on ImageNet, we employ an adversarial loss [9] to increase the realness of the reconstructions. We train a discriminator by minimizing this adversarial loss; therefore, the reconstruction loss becomes:

$$\mathcal{L}_{recon} = \text{MSE}(x, x_{recon}) + \gamma\text{Adversarial}(x, x_{recon}) \ , \tag{5}$$

where $\gamma$ balances the production of realistic reconstructions while maintaining pixel-wise accuracy.

Our major extension to the deformable AEs is the use of a dual-deformation strategy, in which we employ an unconstrained deformer in addition to the constrained deformer, with the aim of improving the localization of atrophic regions. As previously stated, the constrained deformer is trained with a high value of $\beta$ to improve the generation of the pseudo-healthy reconstructions and thus reduce false positives in the anomaly maps. In contrast, the unconstrained deformer has the goal of reverting the pseudo-healthy reconstruction back to its original anomalous state. The deformer is trained with the same loss as in Eq. 3, but with

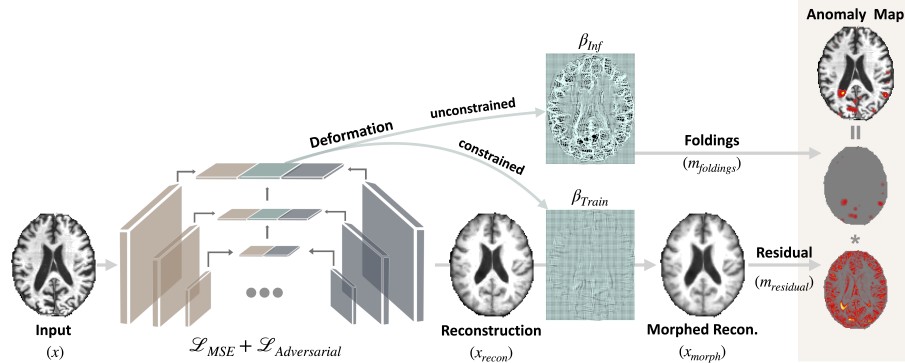

Fig. 1: Our approach, *MORPHADE*, integrates a dual-deformation strategy with a 3D autoencoder and adversarial training. The constrained deformer refines the reconstruction to generate a residual map with reduced false positives, while the unconstrained deformer is used to produce a folding map that highlights anomalies. The residual and folding maps together produce an anomaly map that allows the localization and assessment of the severity of atrophy.

a low value of $\beta$, which allows the creation of unconstrained deformation fields. In such deformation fields, low values of deformation should occur in areas of healthy tissue. Conversely, in regions of atrophy, the deformation field exhibits foldings, or areas in which the mapping of the deformation from the pseudo-healthy reconstruction to the original image is not one-to-one due to the loss of tissue volume. The determinant of the Jacobian of the deformation map, $J_{\boldsymbol{\Phi}}$, can be used to determine local volume changes, with negative values indicating such foldings. Therefore, we highlight the anomalies by using the negative Jacobian values to generate a map of the foldings, $m_{foldings} = \max(0, -\det(J_{\boldsymbol{\Phi}}))$.

We finally multiply these foldings pixel-wise with the residual map from the constrained deformer to generate an anomaly map with reduced false positives and improved atrophy localization:

$$\text{Anomaly Map} = m_{residual} \times m_{foldings} \ . \tag{6}$$

**Implementation.** All networks were trained with Adam optimizer. The discriminator was trained with a learning rate of $1.0e^{-4}$, otherwise $5.0e^{-4}$ was used. The framework was first trained with a high value of $\beta = 10$. We motivate this choice in Fig. 2a, where we show that using decreasing values of $\beta$ during training results in blurrier reconstructions. Conversely, a high $\beta$ value ensures that the AE does not overly rely on the deformations to achieve faithful reconstructions, but is instead forced to learn an accurate representation of the in-distribution data. After 200 epochs, the weights of these models were kept frozen while the deformation parameters were optimized for 100 epochs.

At inference, we use a high value of $\beta = 10$ to obtain the residual maps and a low value of $\beta = 0.01$ to generate the folding maps. We demonstrate the

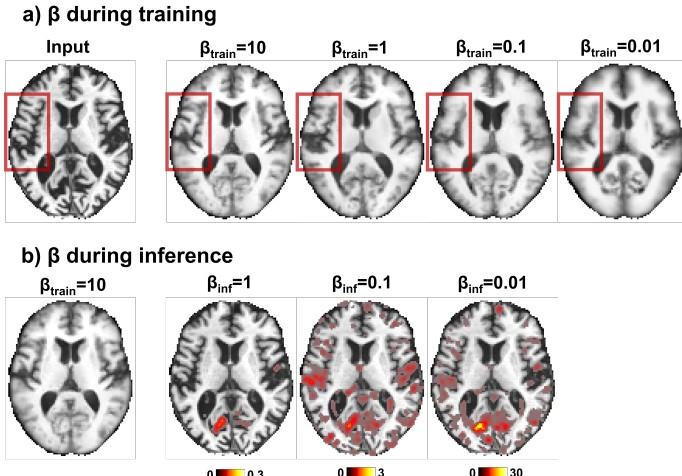

Fig. 2: a) During training, a high value of $\beta = 10$ constrains the deformer, promoting the AE to learn to produce less blurry reconstructions. b) At inference, a lower value of $\beta = 0.01$ is used to generate folding maps (here shown overlayed on the input brain) that enhance the identification of anomalies.

need for lower $\beta$ values to produce improved folding maps in Fig. 2b, where it can be seen that using low values accentuates the anomalous regions in the brain.

**Dataset and Preprocessing.** Data used in the preparation of this article were obtained from the Alzheimer's Disease Neuroimaging Initiative (ADNI) database (adni.loni.usc.edu) [19].We used skull-stripped T1-weighted MPRAGE images of both male and female patients that are registered to the MNI brain template [17]. Our training set comprised 760 healthy control (HC) samples, with an additional 95 HC samples utilized for validation purposes. For the supervised baseline training, an additional 430 AD samples were used. The test set included 215 HC samples and 200 samples with AD.

## 4    Experiments and Results

**Atrophy Localization.** We first validate the effectiveness of our method in identifying atrophy in sub-cortical brain regions affected by AD. To achieve this, we used the FSL FIRST tool [18] to segment these regions and compute mean anomaly scores for each, shown in Fig. 3. Our results indicate that AD patients exhibit notably higher anomaly scores in the hippocampus (left: $0.282 \pm 0.495$, right: $0.185 \pm 0.382$) and amygdala (left: $0.132 \pm 0.207$, right: $0.108 \pm 0.208$) compared to the hippocampus (left: $0.108 \pm 0.193$, right: $0.069 \pm 0.125$) and amygdala (left: $0.066 \pm 0.115$, right: $0.072 \pm 0.111$) for the healthy controls. These results are in line with the clinical expectation of these regions being sig-

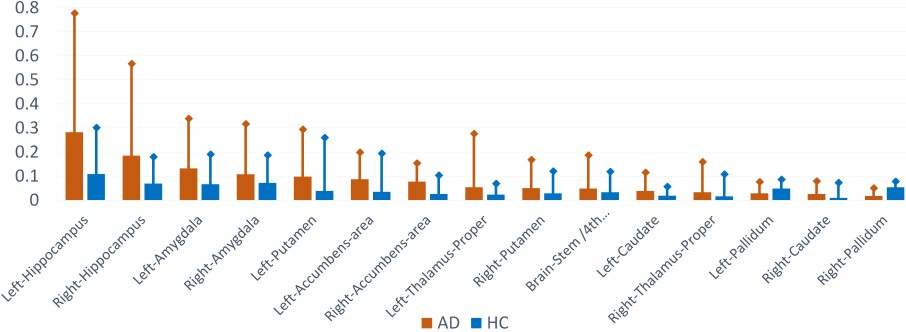

Fig. 3: Anomaly scores for subcortical brain regions for Alzheimer's Disease (AD) and Healthy Control (HC) samples, showcasing markedly higher scores for AD samples in the hippocampus and amygdala, consistent with clinical literature. [6]

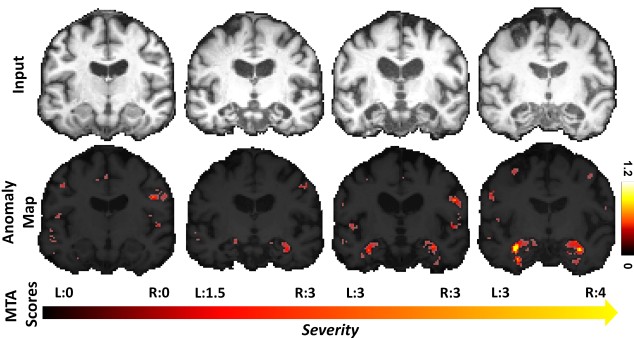

Fig. 4: Anomaly maps for AD patients alongside their corresponding medial temporal lobe atrophy (MTA) scores, demonstrating consistent alignment with AD-related structural changes and clinical MTA assessments.

nificantly affected by AD pathology [6], indicating that *MORPHADE* is able to identify atrophy in clinically relevant brain regions.

**Atrophy Severity.** We next evaluate the ability of our method to determine the severity of the localized anomalies by comparing our anomaly maps to medial temporal lobe atrophy (MTA) scores [20] that were assessed by a senior board-certified neuroradiologist. These scores range from 0 to 4 and are assigned based on the degree of structural changes observed in the choroid fissure, the temporal horn of the lateral ventricle, and the hippocampus. Fig. 4 shows a visual correlation between the degree of atrophy highlighted in the anomaly map in these key regions and the MTA scores, demonstrating the utility of our method in determining the severity of detected anomalies.

Table 1: AUROC scores for the classification of AD and Healthy Controls (HC) patients. Best results are shown in **bold**.

| Method | AD vs. HC ↑ |
|---|---|
| ResNet (Supervised)[14] | 0.77 |
| DenseNet (Supervised)[10] | 0.74 |
| Brainomaly [22] (Mixed Supervision) | 0.78 |
| f-AnoGAN [21] (Unsupervised) | 0.70 |
| Ganomaly [2] (Unsupervised) | 0.72 |
| Adversarial AE (Unsupervised) | 0.74 |
| MORPHADE (ours) (Unsupervised) | **0.80** |
| - Only with residual maps ($\beta$=10) | 0.77 |
| - Only with folding maps ($\beta$=0.01) | **0.79** |

**Pathology Detection.** In this section, we assess the capability of *MORPHADE* in detecting AD at the patient level. Table 1 shows the Area Under the Receiver Operating Characteristic curve (AUROC) scores obtained when comparing our method to various baselines for identifying subjects with AD compared to healthy control (HC) subjects. Our model achieves an AUROC of 0.80, surpassing even the 3D supervised baselines ResNet [14] and DenseNet [10], with AUROCs of 0.77 and 0.74, respectively.

Furthermore, we obtain improved performance compared to methods proposed for unsupervised anomaly detection. These methods are only available in 2D, so were assessed slice-wise with the final anomaly scores obtained by averaging over the slices for each patient. f-AnoGAN [21], Ganomaly [2] obtained AUROCs of 0.70 and 0.72, respectively. We also outperform Brainomaly [22] (AUROC 0.78), a method that is not strictly unsupervised since it requires pathological samples during training for improved performance.

We also compare our results to a 3D adversarial AE to illustrate the benefit of utilizing the deformation fields with our method. Fig. 5 shows the reconstructions and residual maps obtained for both methods in representative AD and healthy controls (HC) subjects. Our method produces more refined reconstructions compared to the adversarial AE, shown by the improved MAE and SSIM scores. Moreover, the residual maps show fewer false positives for the healthy subject, while accentuating pathological areas for the AD subject. Using these improved residual maps alone for AD detection achieves a superior performance of AUROC 0.77 compared to 0.74 obtained by the adversarial AE.

Finally, we demonstrate the utility of our dual-deformation approach, where AD identification was superior using our method compared to using only the residual maps from the constrained deformer (AUROC 0.77) or the folding maps from the unconstrained deformer (AUROC 0.79). Notably, the high performance of the folding maps underscores their effectiveness in detecting anomalies without relying on image differences between the input and reconstructions.

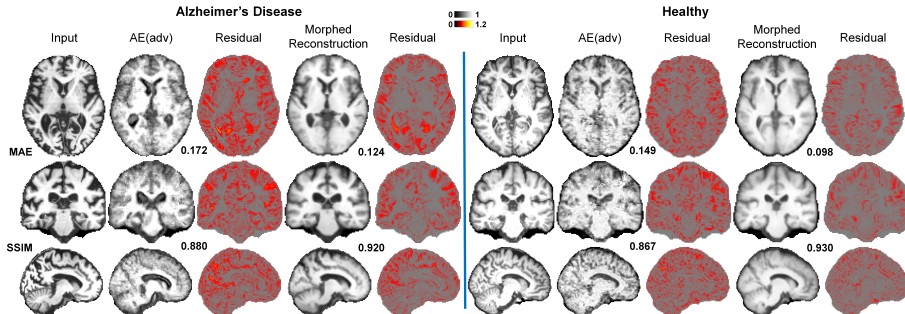

Fig. 5: A comparison of the performance of *MORPHADE* ($\beta = 10$) with adversarial AEs for a subject with AD (left) and a healthy control subject (right). The morphological adjustments facilitated by *MORPHADE* enhance reconstruction fidelity, yielding higher Structure Similarity Index (SSIM) values for our method's morphed reconstructions compared to those of the adversarial AE. The residual maps also demonstrate fewer reconstruction errors for the healthy subject, while highlighting atrophy for the subject with AD.

## 5   Discussion and Conclusion

In this work, we introduced MORPHADE, a novel framework leveraging 3D deformable AEs for unsupervised analysis of Alzheimer's Disease using T1-weighted brain MRI. Our approach is unique in employing deformation fields within an unsupervised learning context to analyze, localize, and assess the severity of AD-related atrophy.

Our results demonstrate that MORPHADE can effectively identify and localize atrophy in clinically relevant brain regions, such as the hippocampus and amygdala, which aligns with clinical expectations of AD pathology. Furthermore, the anomaly maps generated by our method show strong visual correspondence with MTA scores, underscoring the potential of our method in clinical assessments. Lastly, MORPHADE achieved an AUROC of 0.80 in detecting AD, outperforming several supervised and unsupervised baselines. This highlights the robustness of our method without requiring extensive labeled datasets, addressing a significant limitation in current diagnostic approaches.

Future work could explore integrating MORPHADE's deformation metrics with established AD biomarkers, such as tau protein accumulation and amyloid-beta levels, to enhance understanding of disease progression. Additionally, expanding our framework to other neurodegenerative diseases could further validate its versatility and clinical utility.

In conclusion, MORPHADE offers a promising tool for localization, and severity assessment of AD-related atrophy, contributing valuable insights into the progression and diagnosis of neurodegenerative diseases. Our findings suggest that this approach could significantly enhance the non-invasive monitoring and understanding of AD, paving the way for improved patient outcomes.

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
