# OpenReview forum: "Unsupervised Analysis of  Alzheimer’s Disease Signatures using 3D Deformable Autoencoders"
_MICCAI.org/2024/Workshop/MSB — MICCAI Student Board EMERGE Workshop 2024 Oral_

### Official Review · Reviewer_jN6Y · 2024-07-07

**Recommendation:** 5
**Confidence:** 3

**Clarity:**

The paper is clear and well-written, with minor areas for improvement in clarity

**Feedback:**

Even for unsupervised baselines like results using only residual maps is comparable or better than supervised baselines, more explanation can be added here to explain why the method is so good or why supervised methods fail.

**Justification:**

The idea is interesting, method is novel and well explained, both quantitative and qualitative results are shown. The paper qualifies a workshop publication.

**Reproducibility:**

Sufficient amount of details available for reproducing the main results, and open access is provided (or promised upon acceptance) to source code and/or data

**Strengths:**

1. The topic of detecting AD as anomaly is interesting
2. The model is in 3D
3. The paper is well-written, the structure is clear
4. code is public, easy to follow the work
5. the method does not rely on image difference between input and reconstruction to detect anomaly, which is novel and effective

**Summary:**

This paper uses deformation fields in unsupervised AD changes detection. This method is in 3D and adversarial training is introduced in the training process.

**Weaknesses:**

More baseline methods can be compared. Improvements from baselines are marginal.

---

> ### Author Response · Authors · 2024-07-13
> **Rebuttal by Authors**
>
> We thank the reviewer for the thoughtful and constructive feedback and your recognition of novelty and clarity of our work.
>
> Our primary focus and contribution lie in the accurate localization and severity assessment of Alzheimer’s Disease. Furthermore, we have compared our approach with several supervised and unsupervised methods in terms of classification performance. However, we may extend our experiments in future work to include more state-of-the-art methods, appreciating any suggested comparisons from the reviewer.
>
> Our detection results demonstrate improvements even over supervised methods; however, our primary focus extends beyond mere classification accuracy. We aim to localize and assess the severity of structural brain changes due to Alzheimer's Disease through an innovative use of deformations. Our approach generates detailed anomaly maps, providing valuable insights into affected brain regions. It is also important to note that we do not train the networks for classification directly which makes them less susceptible to shortcut learning (as opposed to supervised approaches) and might therefore show improved detection. However, we are eager to explore further aspects of performance, interpretability, early detection, and the clinical impact on patients in future work.

---

### Official Review · Reviewer_1nXW · 2024-07-08

**Recommendation:** 4
**Confidence:** 3

**Clarity:**

The paper is clear and well-written, with minor areas for improvement in clarity

**Feedback:**

Overall, the paper is well-written and contains sufficient description of the proposed methodology. Weaknesses mostly focus on extending the experimentation to more datasets and showcasing the methods' superiority across tasks. Please see also the weakness section for more feedback. Overall, I'm pleased with the quality of the paper and would also raise the score during the rebuttal if the weaknesses are addressed.

Further minor general feedback:

1. The current paper includes statements that would benefit from revision, e.g., in the 2nd paragraph of the conclusion, the authors mention, "This highlights the robustness [...]". However, outperforming competing approaches in one scenario does not indicate robustness. Authors would need to perform dedicated experiments to evaluate robustness.

**Justification:**

Three main weaknesses identified in the paper (i.e., lack of significance tests, lack of diverse experiments, and discrepancy between reported results and prior work) can be addressed in the authors' rebuttal to raise the score but justify the current assignment.

**Reproducibility:**

Sufficient amount of details available for reproducing the main results, but open access is not provided to source code and/or data

**Strengths:**

1. The idea of 3D deformable autoencoders combined with the dual-deformation strategy is novel and shows superior performance compared to recent methods.
2. The correlation between anomaly maps and medial temporal lobe atrophy scores assessed by a clinical expert demonstrates the potential clinical utility of MORPHADE.
3. The availability of anonymized code makes the evaluation transparent.

**Summary:**

This paper introduces MORPHADE, a framework that employs unsupervised learning techniques to detect and localize Alzheimer's Disease in MRI images. The main contribution is the use of 3D deformable autoencoders and a dual-deformation strategy to generate anomaly maps.

**Weaknesses:**

1. The experiments do not include any information about statistical significance. For example, in Table 1, both MORPHADE and MORPHADE with only folding maps are bold (which are described to indicate "best results"), but in the last paragraph of page 7, the authors mention that the results from Table 1 indicate superiority from the combined approach, which is contrary to the presentation. Can the authors thus explain if any statistical tests were performed or which statement is correct?
2. Discrepancies between results of competing methods and original publication, e.g., brainomaly performance in the original paper [1], are significantly lower than in this paper, although they seem to use the same benchmark. Can the authors explain this difference?
3. The paper would benefit from experiments with an openly available dataset to ensure the reproducibility of their evaluation and a broader comparison of their methodology.

Reference:
[1] Siddiquee, M. M., Shah, J., Wu, T., Chong, C., Schwedt, T. J., Dumkrieger, G., Nikolova, S., & Li, B. (2023). Brainomaly: Unsupervised Neurologic Disease Detection Utilizing Unannotated T1-weighted Brain MR Images.

---

> ### Author Response · Authors · 2024-07-13
> **Rebuttal by Authors**
>
> We thank the reviewer for the detailed and constructive feedback and appreciate the positive comments regarding the novelty of our work, and the positive correlation with clinical assessments.
> 1. By presenting both “MORPHADE” and “MORPHADE with only folding maps” in bold in Table 1, we aimed to emphasize the fact that while MORPHADE itself (combining residual maps and folding maps together) achieves the best performance, using only the folding maps alone also demonstrated strong results. This highlights the effectiveness of using deformations for detecting anomalies, without relying on differences between input and reconstruction. While we did not perform statistical significance tests, we will improve the clarity of our results, in particular using a different way to emphasize the results of "Only with folding maps".
> 2. Discrepancies between results: We tried our best to replicate the methods from their original implementations. We hypothesize that variations in the subset and sample sizes may result in performance differences.
> 3. More experiments: Our experiments aimed to demonstrate the efficacy of our method in localizing brain atrophy associated with Alzheimer’s Disease. We analyzed highlighted regions that aligned well with clinical expectations. Additionally, we compared our approach with clinically established MTA scores, evaluated by an expert neuroradiologist, underscoring our method's potential for precise localization and disease severity assessment. Furthermore, we showcased our approach surpassing several supervised and unsupervised baseline methods in terms of classification performance. We welcome suggestions for additional experiments to further extend our findings in future research. Your input would be greatly appreciated.
>
> The ADNI database is already publicly accessible and can be utilized for reproducibility and further research. We have also released a complete csv file with the different subject and scan ids used in our study. We plan to extend our experiments to include more datasets and different pathologies in future work.
>
> We appreciate the suggestion to revise our statement regarding robustness. We agree with the reviewer and will revise the statement in the final version.

---

### Official Review · Reviewer_v5TW · 2024-07-09

**Recommendation:** 5
**Confidence:** 4

**Clarity:**

The paper is clear and well-written, with minor areas for improvement in clarity

**Feedback:**

Overall, I am happy with the paper. It would be a good idea to clarify some things (see first couple of points in the weakness section). For the journal extension, the authors may want to experiment with local pathology like MS lesions and show the usefulness of their work. Similarly, authors may want to utilize hypernetworks to control the effect of hyperparameters like $\beta$.

**Justification:**

Overall, paper is well written, with clear motivation, experiments and results. It would be a good contribution to the workshop.

**Reproducibility:**

Sufficient amount of details available for reproducing the main results, and open access is provided (or promised upon acceptance) to source code and/or data

**Strengths:**

* Clear writing with an explicit explanation of the contributions of the proposed method compared to the previous method (deformable AEs).
* Comparison against many state-of-the-art anomaly detection methods.
* Experimental results show clear performance improvement with the proposed method.
* The use of clinical biomarkers to show the usefulness of the proposed method is commendable.

**Summary:**

The paper proposes to deformation with autoencoders for unsupervised learning to detect AD and localize atrophy in brain MRI.

**Weaknesses:**

* It is not clear how different values of $\beta$ can be used during inference. According to Eq. 3, $\beta$ is a part of the training loss function, and used during inference.
* It is not clear how $m_{foldings}$ was normalized and used in Eq. 6.
* Maybe it would be a good idea to provide statistical significance test with folded validation.

---

> ### Author Response · Authors · 2024-07-13
> **Rebuttal by Authors**
>
> We would like to thank the reviewer for the positive evaluation and the constructive feedback.
>
> We acknowledge the need for further clarity regarding the use of different $\beta$ values. In our final version, we will revise the explanation in the last paragraph of page 4 and will change  "$\beta_{train}$" to  "$\beta_{constrained}$", and "$\beta_{inference}$" to "$\beta_{unconstrained}$" where necessary. Briefly, $\beta$ is used only during training in two phases:
>
> **1. High β Phase:** We train the autoencoder (AE) and constrained deformer with a high $\beta$ to reduce blurriness in the AE reconstructions.
>
> **2. Low β Phase:** We then freeze the AE weights and only train a second deformer with a smaller $\beta$, which we call ‘unconstrained’. The unconstrained deformer generates folding maps during inference.
>
> **3. Inference:** Finally, we perform inference with both the constrained (for obtaining residual maps) and unconstrained (for obtaining the folding maps) deformers.
>
> Regarding the folding maps, $m_{foldings}$, in Equation 6: We use $m_{foldings}$ directly without normalization, however, we use gaussian filtering to smooth them. We will incorporate this into the revised version.
>
> Additionally, we greatly appreciate the reviewer’s suggestions for future work. Adding statistical tests and experimenting with local pathology, such as MS lesions, will be a valuable extension to demonstrate the effectiveness and broader applicability of our method. Furthermore, exploring the use of hypernetworks to control the effect of hyperparameters like $\beta$ is an excellent idea that we plan to explore in future studies.

---

### Official Review · Reviewer_2LP1 · 2024-07-10

**Recommendation:** 5
**Confidence:** 4

**Clarity:**

The paper is clear and well-written, with minor areas for improvement in clarity

**Feedback:**

1. Expanded methodology details:
A. Provide a more in-depth explanation of the dual-deformation strategy, including the rationale behind using different β values for constrained and unconstrained deformers.
B. Include pseudocode or a detailed algorithm description to enhance reproducibility.
C. Elaborate on the choice of adversarial loss and how it compares to perceptual loss in this context.


2. Comprehensive ablation study:
A.Conduct a thorough ablation study to quantify the contribution of each component (3D AE, adversarial training, constrained deformer, unconstrained deformer) to the overall performance.
B. Explore a wider range of β values and their impact on both reconstruction quality and anomaly detection performance.


3. Extended validation:
A. Include additional datasets beyond ADNI to demonstrate generalizability.
B. Perform cross-validation to ensure robustness of results.
C. Consider including mild cognitive impairment (MCI) cases to assess the method's sensitivity to early-stage pathology.


4. Longitudinal analysis:
A. Investigate the method's ability to track disease progression over time using longitudinal data.
B. Assess whether the anomaly scores correlate with cognitive decline measures.

**Justification:**

My recommendation is based on several key factors:  Novelty: MORPHADE presents a unique approach combining 3D deformable autoencoders with a dual-deformation strategy for unsupervised AD analysis. This addresses limitations of both supervised approaches and previous unsupervised techniques. Strong Performance: The method achieves an AUROC of 0.80 in AD detection, outperforming both supervised and unsupervised baselines. This demonstrates its effectiveness and potential to advance the state-of-the-art. Clinical Relevance: The framework generates interpretable anomaly maps correlating with clinical MTA scores, enhancing potential for practical application in healthcare settings. Comprehensive Evaluation: The authors validate their method through multiple experiments, including atrophy localization, severity assessment, and AD detection, strengthening the credibility of the results. Unsupervised Approach: By not requiring labeled training data, MORPHADE addresses a significant challenge in medical image analysis, potentially enabling more widespread and efficient application of AI in AD research and diagnosis. Technical Soundness: The methodology is well-grounded in existing literature while introducing novel elements. The use of adversarial training and dual-deformation strategy are well-justified and effectively implemented.  While there are areas for improvement (e.g., more detailed ablation studies, longitudinal analysis, broader dataset validation), these do not significantly detract from the paper's overall contribution. Instead, they represent opportunities to further strengthen an already solid piece of research. The combination of novelty, strong performance, clinical relevance, and comprehensive evaluation makes this paper a valuable contribution to the field of AD research and medical image analysis. It not only advances the technical state-of-the-art but also has clear potential for real-world impact in improving AD detection and monitoring.

**Reproducibility:**

Sufficient amount of details available for reproducing the main results, and open access is provided (or promised upon acceptance) to source code and/or data

**Strengths:**

1. Novel approach: The use of deformation fields in an unsupervised deep learning framework for AD analysis is innovative and addresses limitations of previous methods.
2. Good performance: MORPHADE outperforms several supervised and unsupervised baselines in AD detection, achieving an impressive AUROC of 0.80.
3. Clinical relevance: The method's ability to localize atrophy in clinically important regions and its correlation with expert-assessed MTA scores highlight its potential for clinical applications.
4. Interpretability: The generation of visual anomaly maps provides interpretable results, which is crucial for clinical adoption and trust in the system.

**Summary:**

This paper introduces MORPHADE, a novel unsupervised framework for analyzing Alzheimer's Disease using 3D brain MRI. The method employs deformable autoencoders with a dual-deformation strategy to generate patient-specific anomaly maps without labeled data. MORPHADE accurately identifies AD-affected brain regions, correlates with clinical assessments, and outperforms both supervised and unsupervised baselines in AD detection. The approach offers a promising tool for improved AD understanding, monitoring, and detection, potentially enabling more efficient and accurate diagnoses.

**Weaknesses:**

1. Limited dataset: While the ADNI dataset is widely used, validating on additional datasets would strengthen the generalizability claims.
2. Computational requirements: The paper doesn't discuss the computational resources needed for training and inference, which could be important for practical implementation.
3. Limited comparison to other unsupervised methods: While some unsupervised baselines are included, a more comprehensive comparison to state-of-the-art unsupervised AD detection methods would be beneficial.

---

> ### Author Response · Authors · 2024-07-13
> **Rebuttal by Authors**
>
> We thank the reviewer for the thorough and insightful feedback and for acknowledging the novelty, strong performance, clinical relevance and comprehensive evaluation of our work.
>
> We conducted training and inference using an NVIDIA RTX A6000 GPU with 48GB of memory. The inference time for a single 3D volume using a plain autoencoder is ~10 ms. Our method is computationally efficient, with an inference time of ~60 ms; the additional use of deformation enables the analysis of atrophy without significantly increasing computational burden. We will include this information in the final version, if space permits.
>
> We showcased our method's effectiveness on the ADNI dataset, which is widely recognized and utilized for Alzheimer’s Disease research. The MICCAI guidelines prevent further experiments, however, we plan to extend our experiments to include more open-source datasets and different pathologies in future work.
>
> We are also grateful for the recommendations for future work. Our primary focus and contribution lie in the accurate localization and severity assessment of Alzheimer’s Disease and we have compared our approach with several supervised and unsupervised methods in terms of classification performance. However, we may extend our experiments in the future to include more state-of-the-art methods, appreciating any suggested comparisons from the reviewer. We will also consider expanding the ablation studies and conducting further analysis longitudinally, as well as with subjects with mild cognitive impairment in future work.

---

### Meta-Review · Area_Chair_iUKv · 2024-07-15

**Recommendation:** Accept (Oral)
**Confidence:** 5

**Metareview:**

The paper is a highly commendable piece of research, showcasing a novel and effective method for unsupervised Alzheimer's Disease detection using 3D brain MRI. The introduction of the dual-deformation strategy and deformable autoencoders represents a significant advancement in the field. The authors have comprehensively addressed all minor concerns the reviewers raised, demonstrating a commitment to clarity and robustness in their methodology. Given the strong experimental results, clinical relevance, and thorough responses to reviewer feedback, this paper is well-rounded and founded.

---

### Decision · Program_Chairs · 2024-07-16

**Decision:**

Accept (Oral)

**Comment:**

This work on unsupervised Alzheimer's disease detection is accepted with minor revisions addressed (as per reviewer feedback). All reviewers found the paper interesting and novel.